# Small conductance $Ca^{2+}$-activated $K^+$ channels induce the firing pause periods during the activation of *Drosophila* nociceptive neurons

Koun Onodera[1], Shumpei Baba[1], Akira Murakami[2,3], Tadashi Uemura[1], Tadao Usui[1]*

[1]Graduate School of Biostudies, Kyoto University, Kyoto, Japan; [2]Faculty of Science, Kyoto University, Kyoto, Japan; [3]Graduate School of Informatics, Kyoto University, Kyoto, Japan

**Abstract** In *Drosophila* larvae, Class IV sensory neurons respond to noxious thermal stimuli and provoke heat avoidance behavior. Previously, we showed that the activated neurons displayed characteristic fluctuations of firing rates, which consisted of repetitive high-frequency spike trains and subsequent pause periods, and we proposed that the firing rate fluctuations enhanced the heat avoidance (Terada et al., 2016). Here, we further substantiate this idea by showing that the pause periods and the frequency of fluctuations are regulated by small conductance $Ca^{2+}$-activated $K^+$ (SK) channels, and the *SK* knockdown larvae display faster heat avoidance than control larvae. The regulatory mechanism of the fluctuations in the Class IV neurons resembles that in mammalian Purkinje cells, which display complex spikes. Furthermore, our results suggest that such fluctuation coding in Class IV neurons is required to convert noxious thermal inputs into effective stereotyped behavior as well as general rate coding.
DOI: https://doi.org/10.7554/eLife.29754.001

*For correspondence:
tadao.usui@gmail.com

Competing interests: The authors declare that no competing interests exist.

## Introduction

Animals sense diverse environmental inputs, including noxious ones, by using specific sensory organs. In principle, sensory neurons convert the intensity of stimuli into the magnitude of firing rates upon sensory transduction (*Adrian, 1926*). For instance, mammalian C-fiber nociceptors convert gentle touch stimuli into relatively low firing rates, whereas injurious forces elicit higher rates (*Delmas et al., 2011*). The 'rate coding' is valuable for sensory transduction, particularly with regard to stimulus intensity; however, the firing rate has an intrinsic upper limit because interspike intervals (ISIs) cannot be shorter than refractory periods, when the membrane is unable to respond to another stimulus (*Berry and Meister, 1998*; *Hodgkin and Huxley, 1952*). This implies that firing rates should saturate at high intensities, at which point the sensory inputs are no longer converted properly in an intensity-to-firing rate correspondence. Therefore, we assume that some sensory neurons may use other coding mechanisms that are employed in the central nervous system (*Avissar et al., 2013*; *Haddad et al., 2013*).

In *Drosophila* larvae, Class IV dendritic arborization neurons (Class IV neurons) are primary nociceptive neurons that respond to multiple stimuli, including high temperature, strong mechanical force, and short-wavelength light (*Hwang et al., 2007*; *Tracey et al., 2003*; *Xiang et al., 2010*). When the neurons are activated by noxious thermal stimuli, for instance, their sensory transduction provokes heat avoidance behavior where larvae rotate around the long body axis in a corkscrew-like manner. A large number of genes responsible for the neuronal activation were identified by

evaluating behavioral phenotypes and monitoring $Ca^{2+}$ dynamics in mutant strains (*Lee et al., 2005*; *Neely et al., 2011*; *Tracey et al., 2003*; *Zhong et al., 2012*); however, there have been few studies which have investigated the coding mechanism of the nociception by recording electrical activity (*Terada et al., 2016*; *Xiang et al., 2010*).

Previously, we built a measurement system using a 1460 nm infrared (IR) laser as a local heating device (*Figure 1—figure supplement 1A*) and found that Class IV neurons responded to noxious thermal stimuli with evoked characteristic fluctuations of firing rates, which consisted of repetitive high-frequency spike trains and subsequent quiescent periods (*Terada et al., 2016*). The occurrence of such 'burst-and-pause' firing patterns was coordinated with large $Ca^{2+}$ increments over the entire dendritic arbors (designated as dendritic $Ca^{2+}$ transients here) and was mediated by L-type voltage-gated $Ca^{2+}$ channels (VGCCs). Knocking down L-type VGCCs in neurons abolished the burst-and-pause firing patterns, and the knockdown larvae displayed delayed heat avoidance behavior. Therefore, we hypothesized that the burst-and-pause firing patterns should be output signals transducing high intensity stimuli and provoking the robust avoidance behavior. However, the regulatory mechanism of the firing patterns remained unclear because L-type VGCCs produce depolarizing currents but not hyperpolarizing ones, which should underlie 'pause' periods. Here, we show that the pause period and the number of the burst-and-pause firing patterns are regulated by small conductance $Ca^{2+}$-activated $K^+$ (SK) channels, and that *SK* knockdown larvae display relatively fast heat avoidance. Furthermore, we show that one of the downstream neurons dramatically changes the response to two optogenetic activations of the Class IV neurons which have distinct numbers of burst-and-pause firing patterns. These findings strengthen the hypothesis and suggest that the 'fluctuation coding' is required to convert high intensities of noxious thermal stimuli into the robust, appropriate avoidance behavior as well as general rate coding.

## Results

### Dendritic $Ca^{2+}$ transients precede unconventional spikes

To understand the molecular mechanism that generates burst-and-pause firing patterns in response to thermal stimuli, we first examined the temporal relationship with dendritic $Ca^{2+}$ transients, whose occurrence was coordinated with the specific firing patterns in an all-or-none fashion. The temporal relationship between the $Ca^{2+}$ transients and unconventional spikes (USs; *Figure 1A*) was unclear because the temporal resolution of monitoring $Ca^{2+}$ dynamics was 30 Hz in our previous work (*Terada et al., 2016*), which was lower than the minimum frequency required to measure differences between spike timings (100–500 Hz; *Lütcke et al., 2013*). Therefore, we monitored $Ca^{2+}$ dynamics with higher temporal resolution (100 Hz) by using a genetically encoded $Ca^{2+}$ indicator GCaMP5G (*Akerboom et al., 2012*), which was brighter than the ratiometric indicator TN-XXL (*Mank et al., 2008*) as employed in our previous work (*Terada et al., 2016*). We then found that all the dendritic $Ca^{2+}$ transients occurred concurrently with USs (*Figure 1B,C and D*). In contrast, when no US occurred or before the first US occurred, $Ca^{2+}$ transients were never observed. To accurately measure the onset of $Ca^{2+}$ transients with stochastic fluctuations, we used an event detection algorithm based on a Schmitt trigger approach (*Grewe et al., 2010*; *Lütcke et al., 2013*) and fit each transient to exponential curves. We then found that the onset of $Ca^{2+}$ transients preceded first-US timings and that the $Ca^{2+}$ transients with multiple USs were stepwise (*Figure 1E and F*; $\Delta t = -50.3 \pm 9.2$ ms, mean ± s.e.m.). We also found that the peak amplitudes of $Ca^{2+}$ transients displayed a positive linear correlation with the number of USs (*Figure 1G*; $p = 1.0 \times 10^{-11}$, *rho* = 0.82, Spearman's rank correlation test). Furthermore, ISIs were shorter before onsets of pauses (*Figure 1H*; $p < 0.05$, paired-sample *t*-test with Bonferroni correction).

We hypothesized that the $Ca^{2+}$ influx mediated by L-type VGCCs amplifies membrane depolarization, which narrows down ISIs of bursts and also induces subsequent pauses. Therefore, we searched for ion channels responsible for generating inhibitory currents that could hyperpolarize membrane potentials during pause periods. We anticipated that activities of such channels must be regulated by intracellular $Ca^{2+}$ concentration ($[Ca^{2+}]_i$) either directly or indirectly.

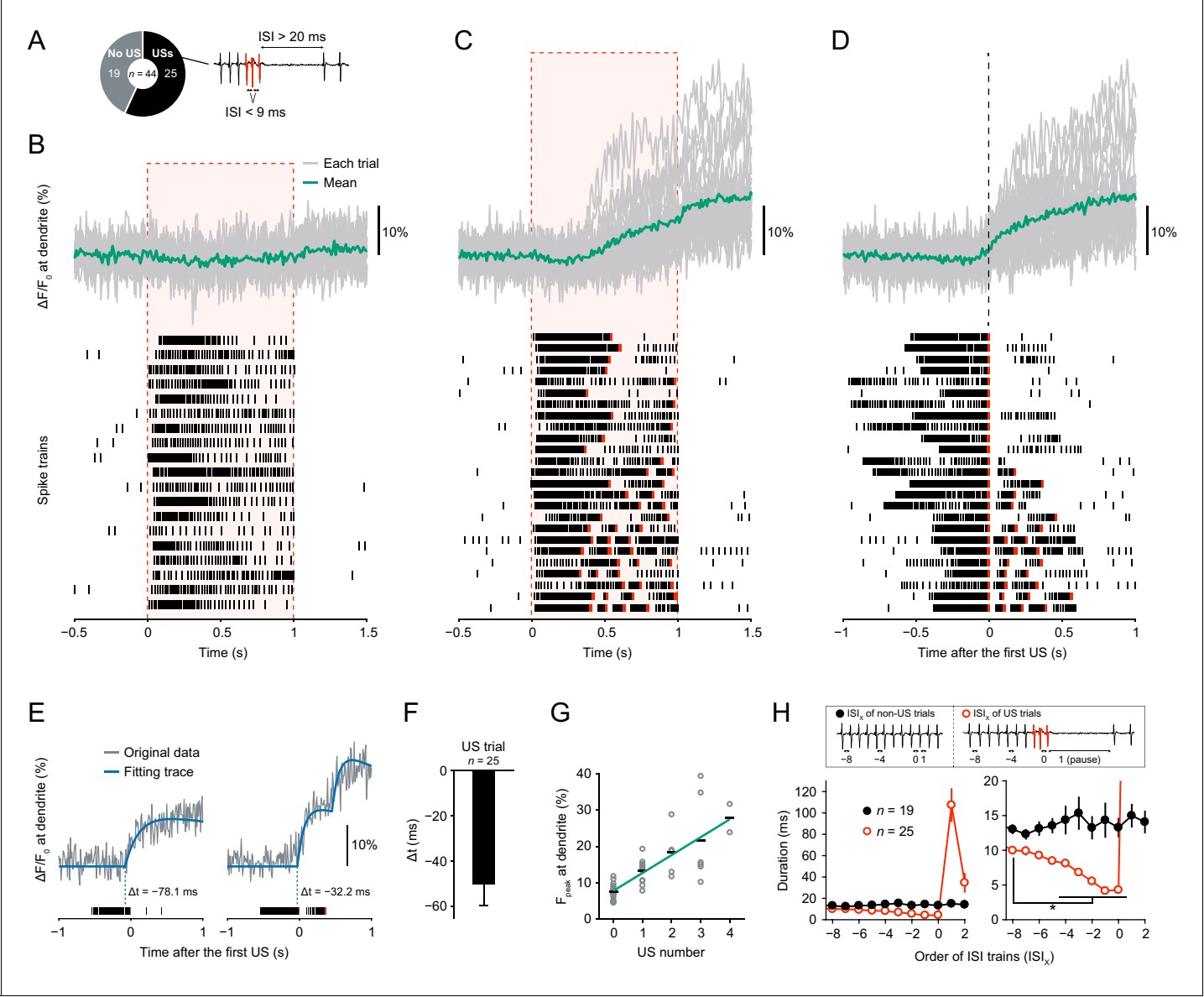

**Figure 1.** Dendritic Ca$^{2+}$ transients precede unconventional spikes. Dual recordings of Ca$^{2+}$ dynamics and extracellular membrane potential in Class IV neurons expressing GCaMP5G. A 44 mW IR laser was focused onto the proximal dendritic arbors in filet preparations for 1 s (red-dashed boxes in B and C). (**A**) Pie chart of recordings (total $n = 44$ cells). To the right, the example illustrates the definition of unconventional spikes (USs, an index of the burst-and-pause firing patterns) as follows: (i) The first and second ISIs of four sequential spikes are less than 9 ms. (ii) The third ISI is longer than 20 ms. Here, sets of the three spikes except for the last one are designated as USs. (**B–D**) Time courses of Ca$^{2+}$ levels at distal dendrites (top) and spike trains (bottom). Data are classified into trials without USs (B) and with USs (C–D). Gray lines indicate dendritic Ca$^{2+}$ transients from each cell, and the green line represents the averaged amplitude. Red raster lines indicate USs. (**B**) Trials without USs did not generate Ca$^{2+}$ transients ($n = 19$ cells; $\Delta F/F_0 = 0.52 \pm 0.87\%$, mean $\pm$ s.e.m. after laser irradiation). (**C**) Trials with USs generated Ca$^{2+}$ transients ($n = 25$ cells; $\Delta F/F_0 = 9.33 \pm 1.57\%$, mean $\pm$ s.e.m. after laser irradiation). The first USs occurred at 0.55 $\pm$ 0.04 s (mean $\pm$ s.e.m.). (D)Data (C) are aligned at the first-US end timings. The onset of the increase in Ca$^{2+}$ levels approximately coincided with the first-US timings. (E)Representative time course of Ca$^{2+}$ transients (gray) and the fitting traces (blue). The onset of Ca$^{2+}$ transients actually preceded the first-US timings, and the Ca$^{2+}$ transients with multiple USs were stepwise (right). (**F**) Temporal differences between the onset of Ca$^{2+}$ transients and the first-US timings. The former occurred earlier than the latter ($\Delta t = -50.3 \pm 9.2$ ms, mean $\pm$ s.e.m.). (**G**) Amplitudes of $F_{peak}$ are plotted against total US numbers for each trial. Short black bars indicate the averages of $F_{peak}$, and the green line is a linear regression of plotted data ($p = 1.0 \times 10^{-11}$, $rho = 0.82$, Spearman's rank correlation test). (**H**) Time course of ISIs. X of ISI$_X$ indicates the order of ISIs: (black) ISI$_{-8}$–ISI$_2$ are the minimum ISI trains of non-US trials in *Figure 1B*. (red) ISI$_0$ indicates the ISIs of the first-US end, and ISI$_1$ represents the pause periods in *Figure 1C*. At the right, the y-axis was magnified to show that ISIs became shorter before the occurrence of the pause (mean $\pm$ s.e.m.; *$p < 0.05$, paired-sample *t*-test with Bonferroni correction).

DOI: https://doi.org/10.7554/eLife.29754.002

*Figure 1 continued on next page*

*Figure 1 continued*

The following source data and figure supplement are available for figure 1:

**Source data 1.** Source data for *Figure 1*.
DOI: https://doi.org/10.7554/eLife.29754.004
**Source data 2.** Source data for *Figure 1—figure supplement 1*.
DOI: https://doi.org/10.7554/eLife.29754.005
**Figure supplement 1.** The recording system and maximum firing rates regarding the US number.
DOI: https://doi.org/10.7554/eLife.29754.003

## Electrophysiological screen of $Cl^-$ channels and $K^+$ channels

To elucidate the regulatory mechanism of the pause, we screened ion channels that might generate hyperpolarizing currents: such channels include outward $K^+$ and inward $Cl^-$ channels. The direction of passive transport of each ion through channels is dependent on the electrochemical gradient across the plasma membrane, but the intracellular $Cl^-$ concentration ($[Cl^-]_i$) is largely different among cells (*Kaila et al., 2014*) and the direction of $Cl^-$ transport in Class IV neurons was unclear. We therefore monitored $Cl^-$ dynamics by a genetically encoded FRET-based $Cl^-$ indicator, Super-Clomeleon (*Grimley et al., 2013*; *Figure 2A*) and found that the FRET ratio increased at both somata and distal dendrites upon IR-laser irradiation (*Figure 2B and C*). Because the FRET ratio of SuperClomeleon rises as the $[Cl^-]_i$ falls, we expected that the $[Cl^-]_i$ should decrease upon stimulation. These results suggest that the passive $Cl^-$ transport in Class IV neurons is outward and generates depolarizing currents but not hyperpolarizing ones. In parallel, we investigated the role of one of the $Ca^{2+}$-activated $Cl^-$ channels, Subdued (*Jang et al., 2015*), and showed that it can contribute to membrane excitation in the neurons (*Figure 2—figure supplement 1*). Thus, we excluded $Cl^-$ channels as candidate sources of hyperpolarizing currents.

Next, we focused on the roles of various $K^+$ channels as mediators of hyperpolarization. The *Drosophila melanogaster* genome has 29 genes that encode pore-forming subunits of $K^+$ channels, including 11 voltage-gated $K^+$ channels (*Sh*, *eag*, etc.), 11 two-pore domain $K^+$ channels (*Task6*, *sand*, etc.; *Pimentel et al., 2016*) and others (*Figure 2—source data 1*). We first knocked down each of the candidate genes in Class IV neurons and recorded electrical activities of the knockdown neurons upon IR-laser irradiation, and examined the properties of burst-and-pause firing patterns. We found that the number of USs was significantly increased in five different gene knockdown neurons (*Sh*, *Shal*, *SK*, *Irk2*, and *Task7*; *Figure 2D and E*; $p<0.05$, Wilcoxon rank sum test). The five knockdown neurons also exhibited an increased number of 'peaks' (*Figure 2—figure supplement 2*; $p<0.05$, Wilcoxon rank sum test), as defined in our previous study (*Terada et al., 2016*). Furthermore, although knocking down L-type VGCC abolishes dendritic $Ca^{2+}$ transients (*Terada et al., 2016*), the five $K^+$ channel knockdowns did not decrease the amplitudes of the $Ca^{2+}$ transients (*Figure 2F*). These results suggested that these candidates participate in an unknown mechanism downstream of the $Ca^{2+}$ influx. Notably, *SK* encodes a small conductance $Ca^{2+}$-activated $K^+$ channel, which can be activated by dendritic $Ca^{2+}$ influx, and therefore could be one of the major factors underlying hyperpolarization. Thus, we explored how SK channels contributed to shaping the burst-and-pause firing patterns.

## SK channels generate pause periods

To investigate the physiological roles of SK channels, we stimulated the knockdown neurons with different IR-laser powers. We then found that *SK* knockdown increased the firing properties, including the US number, peak number and maximum firing rate, even with low laser powers (*Figure 3A–D*, *Figure 3—figure supplement 1A*). Importantly, the *SK* knockdown shortened the pause periods (*Figure 3E*; median of pause period: [*ppk-Gal4*] 103.9 ms, [*UAS-SK RNAi*] 112.9 ms, [*ppk>SK RNAi*] 46.75 ms; $p<0.001$, Student's *t*-test with Holm correction), which suggested that SK-dependent current regulates the pause period. Similar firing changes were also observed in two additional *SK* knockdown neurons, targeting two different sequences in the *SK* gene (*Figure 3—figure supplement 2A–D*, *Figure 3—figure supplement 1B*). We speculated that SK channels would be dramatically activated by the sudden increase in $[Ca^{2+}]_i$ through L-type VGCCs, and would generate a transient hyperpolarizing $K^+$ current. Nevertheless, it was important to rule out a more trivial

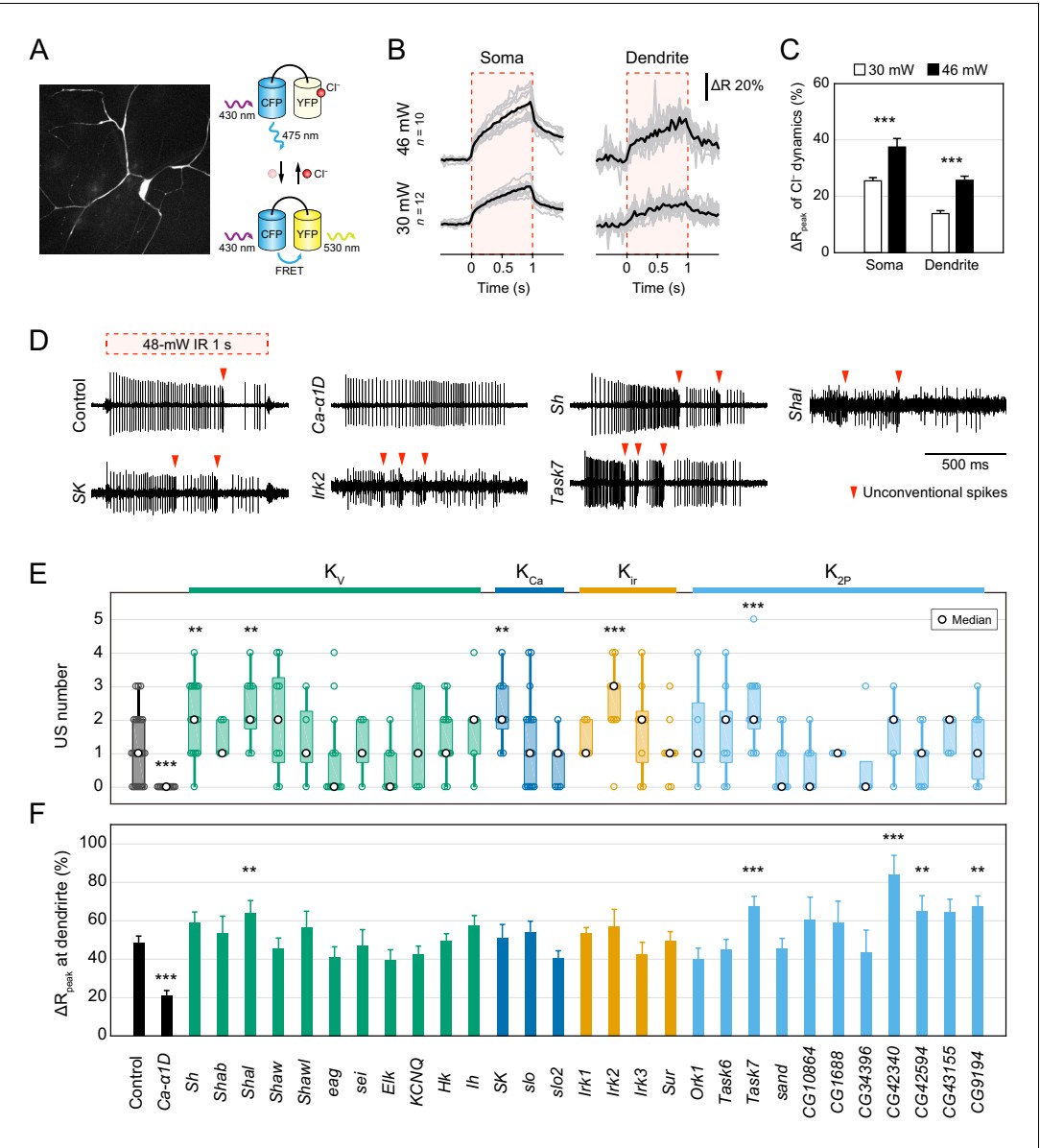

**Figure 2.** Electrophysiological screen of Cl⁻ and K⁺ channels. (**A–C**) Cl⁻ dynamics of Class IV neurons expressing SuperClomeleon. The IR laser (30 and 46 mW) was focused onto the proximal dendritic arbors in whole-mount preparations for 1 s (red-dashed boxes in B). (**A**) A schematic diagram of Cl⁻ indicator SuperClomeleon. The FRET ratio decreases upon an influx of Cl⁻, due to quenching of YFP fluorescence by reversible Cl⁻ binding. Left is a representative CFP image before IR-laser irradiation. (**B**) Time courses of the FRET ratio at somata (left) and distal dendrites (right) in wild-type neurons. Both of them increased upon IR-laser irradiation. Gray lines indicate each of the Cl⁻ changes, and black lines represent the averaged amplitudes. The apparent efflux of Cl⁻ ions was unexpected. (**C**) Amplitudes of $\Delta R_{peak}$ of SuperClomeleon increased with IR-laser power (mean ± s.e.m.; ***p<0.001, Student's $t$-test). (**D–F**) Responses of screened neurons expressing the $Ca^{2+}$ indicator TN-XXL. The 48 mW IR laser was focused onto the proximal dendritic arbors in filet preparations for 1 s (red-dashed box in D). **p<0.05, ***p<0.01 versus control. (**D**) Representative recordings of control, $Ca$-$\alpha 1D$ (L-type VGCC $\alpha_1$ subunit gene) RNAi and K⁺ channel-coding gene (*Shaker*, *Shal*, *SK*, *Irk2* and *Task7*) RNAi neurons. (**E**) Boxplot of the total US number in screened neurons. The US number increased in five different gene knockdown neurons (*Sh*, *Shal*, *SK*, *Irk2* and *Task7*; Wilcoxon rank sum test). (**F**) Amplitudes of the dendritic $Ca^{2+}$ transients in screened channels. The amplitudes did not decrease except for $Ca$-$\alpha 1D$ RNAi neurons (mean ± s.e.m.; Student's $t$-test). Bottom horizontal labels indicate symbols of knocked down genes and upper labels represent channel families: $K_v$, voltage-gated K⁺ channel; $K_{Ca}$, $Ca^{2+}$-activated K⁺ channel; $K_{ir}$, Inward rectifier K⁺ channel; $K_{2P}$, Two-pore domain K⁺ channel.

DOI: https://doi.org/10.7554/eLife.29754.006

*Figure 2 continued on next page*

*Figure 2 continued*

The following source data and figure supplements are available for figure 2:

**Source data 1.** Twenty-Nine K$^+$ channels were screened.
DOI: https://doi.org/10.7554/eLife.29754.009
**Source data 2.** Source data for *Figure 2*.
DOI: https://doi.org/10.7554/eLife.29754.010
**Source data 3.** Source data for *Figure 2—figure supplement 1*.
DOI: https://doi.org/10.7554/eLife.29754.011
**Source data 4.** Source data for *Figure 2—figure supplement 2*.
DOI: https://doi.org/10.7554/eLife.29754.012
**Figure supplement 1.** Behavioral and electrophysiological analysis of *subdued* mutants.
DOI: https://doi.org/10.7554/eLife.29754.007
**Figure supplement 2.** Quantification of the maximum firing rate and the peak number.
DOI: https://doi.org/10.7554/eLife.29754.008

explanation for the changes in firing patterns in the *SK* knockdown neurons, which might possibly be due to altered dendritic architecture. We quantified the dendritic morphology of *SK* knockdown neurons and concluded that the altered physiological responses were not due to morphological defects in the dendritic arbors (*Figure 3—figure supplement 3*).

We hypothesized that the burst-and-pause firing patterns should be output signals provoking robust heat avoidance behavior. To test this hypothesis, we examined how *SK* knockdown larvae responded to thermal stimulation, and we found that they displayed significantly faster onsets of responses; moreover, the response rate was increased upon moderate stimulation (44°C, *Figure 3F and G*; 42°C, *Figure 3—figure supplement 2E F*). These results suggested that the enhanced behavioral responses are induced either by the increment in the US number or by the changes in the other firing properties (pause period and maximum firing rate, etc.). We previously reported that the L-type VGCC knockdown abolished the burst-and-pause firing patterns and provoked a delayed response to thermal stimuli (*Terada et al., 2016*). Consistent with these findings, we propose that the burst-and-pause firing patterns should be output signals provoking the robust avoidance behavior.

We also examined heat avoidance behavior of larvae, where one of the other three candidate genes (*Shal*, *Irk2*, and *Task7*) was knocked down; however, the respective knockdown larvae did not show any difference in the response rate of avoidance (*Figure 3—figure supplement 4A and B*). Interestingly, the frequencies of spontaneous spikes were significantly increased in the three knockdown Class IV neurons but not in *SK* knockdown ones (*Figure 3—figure supplement 4C and D*). Notably, a recent study revealed that the activation of Class IV neurons during larval development inhibited the synaptic transmission to second-order neurons via serotonergic feedback signaling and suppressed the avoidance behavior (*Kaneko et al., 2017*). In addition, the topographic projections of Class IV neurons are partially dependent on the levels of neuronal activity, including spontaneous spikes (*Kaneko and Ye, 2015*; *Yang et al., 2014*). Thus, the elevated basal neuronal activity during development might decrease the efficacy of synaptic transmission and/or remodel synaptic connections of the neurons, which would tend to counteract the effect of the increment of firing rate fluctuations on the avoidance behavior of these knockdown animals.

The downstream circuitry of the Class IV neurons has been identified through functional and anatomical approaches (*Chin and Tracey, 2017*; *Hu et al., 2017*; *Ohyama et al., 2015*; *Vogelstein et al., 2014*; *Yoshino et al., 2017*). To explore any differences in the responses of that circuitry when Class IV neurons evoked various firing patterns, we examined the neuronal activity of Goro neurons in response to optogenetic activations of Class IV neurons (*Figure 3—figure supplement 5A*). We first investigated firing patterns induced by optogenetic activations in Class IV neurons and found that the numbers of burst-and-pause patterns were significantly different between continuous and intermittent illuminations (*Figure 3—figure supplement 5B–D*; p<0.001, Wilcoxon signed-rank sum test). Importantly, the total spike numbers and maximum firing rates were comparable between the two conditions (*Figure 3—figure supplement 5E and F*). We then examined whether the activity of the downstream neurons should be differentially induced with the type of optogenetic manipulation. We found that the maximum amplitude of Ca$^{2+}$ rises in Goro neurons

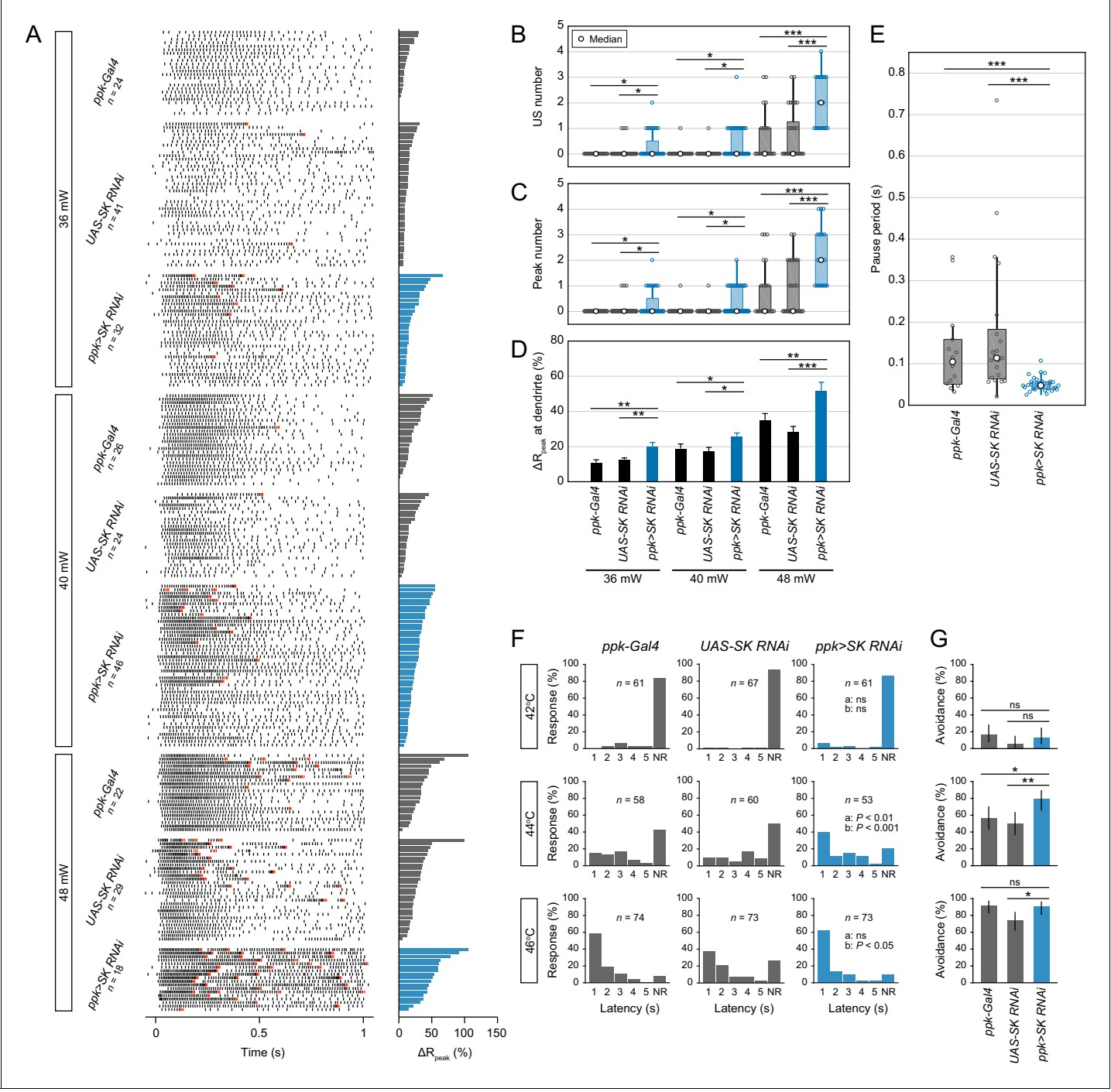

**Figure 3.** SK channels generate pause periods. (A–E) Responses of two control neurons (*ppk-Gal4* and *UAS-SK RNAi*[HMJ21196]) and *SK* knockdown neurons (*ppk>SK RNAi*[HMJ21196]) with different IR-laser power settings (36, 40 and 48 mW). The IR laser was focused onto the proximal dendritic arbors in filet preparations for 1 s. (A)Raster plots of firing (left) and magnitudes of the $\Delta R_{peak}$ corresponding to dendritic $Ca^{2+}$ transients (right). Trials are sorted in descending order of the magnitude of the $\Delta R_{peak}$. Red raster lines indicate USs. (B–D) *SK* knockdown neurons increased the US number, peak number (B and C; boxplots; Wilcoxon rank sum test with Holm correction), and amplitude of the dendritic $Ca^{2+}$ transients (D; mean ± s.e.m.; Student's *t*-test with Holm correction) with three different laser powers. (E) Boxplots of the pause periods triggered by the 48 mW IR laser. Pause periods were shortened in *SK* knockdown neurons (median: [*ppk-Gal4*] 103.9 ms (*n* = 14), [*UAS-SK RNAi*] 112.9 ms (*n* = 21), [*ppk >SK RNAi*] 46.75 ms (*n* = 34); Student's *t*-test with Holm correction). (F–G) Avoidance behavior of two control larvae and *SK* knockdown larvae in response to thermal stimulation (42, 44, and 46°C). (F) The distribution of response latency. *SK* knockdown larvae displayed fast onsets of responses upon moderate stimulation (44°C; median: [*ppk-Gal4*] 3.80 s, [*UAS-SK RNAi*] 4.86 s, [*ppk>SK RNAi*] 1.80 s; Wilcoxon rank sum test with Holm correction). Neither control nor *SK*

*Figure 3 continued on next page*

*Figure 3 continued*

knockdown larvae showed avoidance behavior upon lower stimulation (42°C), whereas most of the larvae displayed it with higher stimulation (46°C). NR, no response group. 'a' is a *P* value versus *ppk-Gal4*, and 'b' is that versus *UAS-SK RNAi*. (G) Percentage of larvae responding within 5 s with 95% Clopper-Pearson confidence intervals. The response rate of *SK* knockdown larvae increased upon moderate stimulation (44°C: [*ppk-Gal4*] 56.9%, [*UAS-SK RNAi*] 50.0%, [*ppk>SK RNAi*] 79.2%; Fisher's exact test with Holm correction). *p<0.05, **p<0.01, ***p<0.001.
DOI: https://doi.org/10.7554/eLife.29754.013

The following source data and figure supplements are available for figure 3:

**Source data 1.** Source data for *Figure 3*.
DOI: https://doi.org/10.7554/eLife.29754.019
**Source data 2.** Source data for *Figure 3—figure supplement 1*.
DOI: https://doi.org/10.7554/eLife.29754.020
**Source data 3.** Source data for *Figure 3—figure supplement 2*.
DOI: https://doi.org/10.7554/eLife.29754.021
**Source data 4.** Source data for *Figure 3—figure supplement 3*.
DOI: https://doi.org/10.7554/eLife.29754.022
**Source data 5.** Source data for *Figure 3—figure supplement 4*.
DOI: https://doi.org/10.7554/eLife.29754.023
**Source data 6.** Source data for *Figure 3—figure supplement 5*.
DOI: https://doi.org/10.7554/eLife.29754.024
**Figure supplement 1.** Maximum firing rates regarding the presence of USs and the effect of *SK* knockdown.
DOI: https://doi.org/10.7554/eLife.29754.014
**Figure supplement 2.** Analysis of two different *SK* knockdowns.
DOI: https://doi.org/10.7554/eLife.29754.015
**Figure supplement 3.** Dendritic morphology of *SK* knockdown Class IV neurons.
DOI: https://doi.org/10.7554/eLife.29754.016
**Figure supplement 4.** Avoidance behavior and spontaneous spikes upon knockdown of the other $K^+$ channels.
DOI: https://doi.org/10.7554/eLife.29754.017
**Figure supplement 5.** Differential $Ca^{2+}$ dynamics of downstream neurons induced by optogenetic activations of Class IV neurons.
DOI: https://doi.org/10.7554/eLife.29754.018

was larger upon activation accompanied with more burst-and-pause firing patterns in Class IV neurons (*Figure 3—figure supplement 5G and H*; [continuous] $F_{peak}$ = 11.3 ± 1.8%, [intermittent] $F_{peak}$ = 19.4 ± 2.2%, mean ± s.e.m.; p<0.01, Welch's *t*-test). The results suggested that firing rate fluctuations were decoded in downstream circuits separately from the firing rate itself. Although the mechanism by which burst-and-pause firing patterns are read out as downstream electrical signals is still unknown, we can address this question by examining the activities of the other neurons in the circuitry.

## Discussion

Although the increased number of USs in *SK* knockdown neurons may initially seem counterintuitive, it can be explained comprehensively by two states of SK channels, at low and high activation levels (*Figure 4A*): (i) Before USs occur, most SK channels are in the steady state because the $Ca^{2+}$/calmodulin association is restricted at low $[Ca^{2+}]_i$, and the SK current slightly inhibits the incidence of firings during burst periods. Therefore, *SK* knockdown attenuates the inhibition of firings, which raises the occurrence rate of USs. (ii) In contrast, after USs occur with dendritic $Ca^{2+}$ transients, the channels are shifted to the activation state by high $[Ca^{2+}]_i$, and the current greatly promotes afterhyperpolarization, which generates the pause periods. Thus, the knockdown dramatically decreases the pause periods, which shortens the time requiring one burst-and-pause firing pattern. Due to the two impacts on firings, the US number per unit time would be expected to increase upon *SK* knockdown.

We hypothesize that the burst-and-pause firing patterns in Class IV neurons are regulated by functional coordination between L-type VGCCs and SK channels as follows (*Figure 4B*): (1) Thermosensitive channels including dTrpA1 and Painless (*Luo et al., 2017*; *Neely et al., 2011*; *Tracey et al., 2003*; *Zhong et al., 2012*) are activated by high-temperature stimulation and elicit the initial membrane depolarization in the dendritic arbors. (2) Once the membrane potential of soma

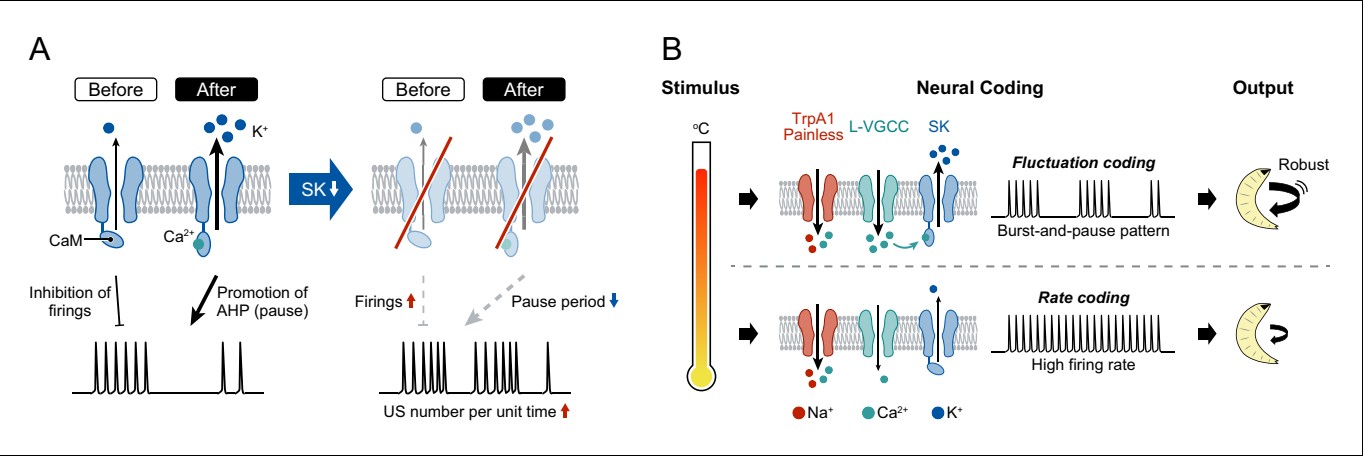

**Figure 4.** A model of information processing. (**A**) Two states of SK channels, and the activation level. Here, calmodulin (CaM, ellipses) is illustrated by tethering to the intracellular terminus of the SK channels. Before USs (or without USs), the small SK current inhibits the incidence of firings during burst periods. After USs, the large SK current promotes after-hyperpolarization (AHP), which induces pause periods. See further explanation in Discussion. (**B**) The regulatory mechanism of burst-and-pause firing patterns by functional coordination between L-type VGCCs and SK channels in Class IV neurons. General rate coding occurs in Class IV neurons at relatively low temperatures; this is mediated by thermoTRPs (TrpA1 and Painless) and many types of voltage-gated ion channels (not illustrated here). At higher temperature, however, L-type VGCCs and SK channels convert the firing from continuous high-frequency patterns into burst-and-pause patterns. This mechanism allows fluctuation coding in sensory neurons.

DOI: https://doi.org/10.7554/eLife.29754.025

The following figure supplement is available for figure 4:

**Figure supplement 1.** Information processing by the burst-and-pause firing pattern.

DOI: https://doi.org/10.7554/eLife.29754.026

exceeds a certain threshold by the prolonged stimulation, the neurons evoke action potentials and then increase firing rates with the intensity of stimulation ('rate coding'). (3) When L-type VGCCs in the dendritic arbors are activated by the high-order depolarization, they induce a large $Ca^{2+}$ influx, which rapidly activates SK channels. (4) The activated SK channels produce a hyperpolarizing current, thereby generating the pause periods ('fluctuation coding'). We also suggest that other $K^+$ channels may slightly contribute to the generation of pauses, because the pause periods were not completely abolished in *SK* knockdown neurons (*Figure 3E*, *Figure 3—figure supplement 2D*). Although the other candidate channels, such as *Sh* and *Shal*, are not activated by the $[Ca^{2+}]_i$ rise, most of them are voltage-dependent and hence hyperpolarize the membrane potential to some degree after depolarization, regardless of dendritic $Ca^{2+}$ influx. Because the hyperpolarization suppresses the probability of firing, including US, the knockdown of those channels should lead to the increment of the US number.

In the mammalian cerebellar cortex, climbing fiber inputs evoke complex spikes of Purkinje cells, which induce a dendritic $Ca^{2+}$ influx through $Ca^{2+}$ spikes and subsequent pauses (*Davie et al., 2008*; *Kitamura and Häusser, 2011*; *Llinás and Sugimori, 1980*; *Mathews et al., 2012*). The pause periods of post-complex spikes are regulated by dendritic $Ca^{2+}$ spikes, which are dependent on P/Q-type VGCCs (*Davie et al., 2008*), and are modulated by after-hyperpolarization, which is largely dependent on SK2 channels (*Grasselli et al., 2016*). Considering these observations, the regulatory mechanism of complex spikes is remarkably similar to that of burst-and-pause firing patterns in Class IV neurons (*Figure 4—figure supplement 1A*).

In principle, sensory neurons convert the intensity of stimuli into the magnitude of firing rates (*Adrian, 1926*). This form of rate coding also occurs in Class IV neurons at relatively low temperatures, and it is mediated by thermosensitive channels and many types of voltage-gated ion channels (*Figure 4B*, *Figure 4—figure supplement 1B and C*). At higher temperature, however, L-type VGCCs and SK channels modulate the firing, transitioning from continuous high-frequency patterns into burst-and-pause patterns. Thus, we propose that the firing-rate-fluctuation coding allows

sensory neurons to transmit strong stimuli not covered in rate coding, thereby provoking robust avoidance behavior.

## Materials and methods

### *Drosophila* mutant and transgenic strains

The transgenic line expressing the FRET-based $Ca^{2+}$ indicator TN-XXL (*Mank et al., 2008*) in Class IV neurons was *3×[ppk-TN-XXL] (attP40)* from our previous work (*Terada et al., 2016*). Mutants of one of the Anoctamin family channels, Subdued, were *subdued$^{\Delta5265}$* and *Df(3R)Exel6184*, from C. Kim. A transgenic line expressing the split Gal4 was *R72F11_AD; R52F07_DBD* from T. Ohyama. A transgenic line expressing the FRET-based $Cl^-$ indicator SuperClomeleon was *UAS-SuperClomeleon* (FBst0059847; *Haynes et al., 2015*) from the Bloomington Stock Center. Transgenic lines expressing channel RNAi were from the Bloomington Stock Center and Vienna *Drosophila* Resource Center (see also *Figure 2—source data 1*). Other transgenic lines were *pickpocket (ppk)-Gal4* (FBst0032078), *20 × UAS-GCaMP5G* (FBst0042037), *UAS-Dcr-2* (FBst0024651), *TrpA1-QF* (FBst0036348), *R69F06-Gal4* (FBst0039497), *10XQUAS-ChR2.T159C-HA* (FBst0052259), and *20XUAS-IVS-NES-jRCaMP1b-p10* (FBst0063793), from the Bloomington Stock Center.

Exact genotypes of individual animals used in figures are described below:

### Figure 1
*ppk-Gal4/20 × UAS-GCaMP5G (attP40)*

### Figure 2
(B–C) *ppk-Gal4/20 × UAS SuperClomeleon (attP40)*
(D–F) *ppk-Gal4, 3×[ppk-TN-XXL] (attP2), UAS-each channel RNAi (see Figure 2—source data 1)*

### Figure 3
*w$^{1118}$; ppk-Gal4/+; 3×[ppk-TN-XXL] (attP2)/+ ('ppk-Gal4')*
*UAS-SK RNAi$^{HMJ21196}$/+; 3×[ppk-TN-XXL] (attP2)/+ ('UAS-SK RNAi')*
*ppk-Gal4/UAS-SK RNAi$^{HMJ21196}$; 3×[ppk-TN-XXL] (attP2)/+ ('ppk>SK RNAi')*

### Figure 1—figure supplement 1
(B) *ppk-Gal4/20 × UAS-GCaMP5G (attP40)*

### Figure 2—figure supplement 1
(A–F) *3×[ppk-TN-XXL] (attP40)/+ ('WT')*
*3×[ppk-TN-XXL] (attP40)/+; subdued$^{\Delta5265}$/Df(3R)Exel6184 ('subdued')*
(G–H) *ppk-Gal4/20 × UAS SuperClomeleon (attP40); subdued$^{\Delta5265}$/Df(3R)Exel6184*

### Figure 2—figure supplement 2
*ppk-Gal4, 3×[ppk-TN-XXL] (attP2), UAS-each channel RNAi (see Figure 2—source data 1)*

### Figure 3—figure supplement 2
*w$^{1118}$; ppk-Gal4/+; UAS-Dcr-2/+ ('Control')*
*w$^{1118}$; ppk-Gal4/+; UAS-Dcr-2/UAS-SK RNAi$^{GD12601}$ ('SK RNAi$^{GD}$')*
*w$^{1118}$; ppk-Gal4/UAS-SK RNAi$^{KK107699}$; UAS-Dcr-2/+ ('SK RNAi$^{KK}$')*

### Figure 3—figure supplement 1
(A) *w$^{1118}$; ppk-Gal4/+; 3×[ppk-TN-XXL] (attP2)/+ ('ppk-Gal4')*
*UAS-SK RNAi$^{HMJ21196}$/+; 3×[ppk-TN-XXL] (attP2)/+ ('UAS-SK RNAi$^{HMJ}$')*
*ppk-Gal4/UAS-SK RNAi$^{HMJ21196}$; 3×[ppk-TN-XXL] (attP2)/+ ('ppk>SK RNAi$^{HMJ}$')*
(B) *w$^{1118}$; ppk-Gal4/+; UAS-Dcr-2/+ ('Control')*
*w$^{1118}$; ppk-Gal4/+; UAS-Dcr-2/UAS-SK RNAi$^{GD12601}$ ('ppk>SK RNAi$^{GD}$')*
*w$^{1118}$; ppk-Gal4/UAS-SK RNAi$^{KK107699}$; UAS-Dcr-2/+ ('ppk>SK RNAi$^{KK}$')*

## Figure 3—figure supplement 3

$w^{1118}$; *ppk-Gal4/+; 3×[ppk-TN-XXL] (attP2)/+ ('ppk-Gal4')*
*UAS-SK RNAi[HMJ21196]/+; 3×[ppk-TN-XXL] (attP2)/+ ('UAS-SK RNAi')*
*ppk-Gal4/UAS-SK RNAi[HMJ21196]; 3×[ppk-TN-XXL] (attP2)/+ ('ppk>SK RNAi')*
*3×[ppk-TN-XXL] (attP2)/UAS-Sur RNAi[GL00506]*
*ppk-Gal4/+; 3×[ppk-TN-XXL] (attP2)/UAS-Sur RNAi[GL00506]*

## Figure 3—figure supplement 4

$w^{1118}$; *ppk-Gal4/+; 3×[ppk-TN-XXL] (attP2)/+ ('ppk-Gal4')*
*ppk-Gal4, 3×[ppk-TN-XXL] (attP2), UAS-each channel RNAi (see Figure 2—source data 1)*

## Figure 3—figure supplement 5

(B–F)
*TrpA1-QF 10XQUAS-ChR2.T159C-HA/R72F11_AD; 20XUAS-IVS-NES-jRCaMP1b-p10/R52F07_AD*
(G–H)
*TrpA1-QF 10XQUAS-ChR2.T159C-HA/+; 20XUAS-IVS-NES-jRCaMP1b-p10/R69F06-Gal4*
*('Class IV>ChR2, Goro>jRCaMP')*

## Electrophysiology and IR-laser irradiation

Preparation of larvae and extracellular recording were performed as previously described (*Terada et al., 2016*, *Figure 1—figure supplement 1A*). The foci of the infrared (IR)-laser irradiation were targeted onto the proximal dendritic arbors, essentially as described in our previous analyses. The time window for the experimental irradiation was 1 s except for data for *Figure 2—figure supplement 1C* (30 mW IR, 5 s) and *Figure 3A* (36-mW IR, 5 s), which were quantified during the initial 1 s. Quantification of the maximum firing rate and the peak number of firing rate fluctuations was performed as previously described (*Terada et al., 2016*) with slight modifications (see *Figure 2—figure supplement 2A*). For quantification of the pause period, we excluded USs that had occurred around the shutdown of IR-laser irradiation and were not accompanied by additional spikes during the irradiation.

## Ca²⁺ imaging

We used a TN-XXL indicator except for *Figure 1* because it allowed more robust quantitative analysis in the presence of perturbations along Z-axis motions by larval body wall muscles. Ca$^{2+}$ imaging of TN-XXL-expressing Class IV neurons was performed as previously described (*Terada et al., 2016*). $\Delta R$ is the change of fluorescence ratio and $\Delta R_{peak}$ is defined as the maximum. Ca$^{2+}$ imaging of GCaMP5G was performed on filet preparations. GCaMP5G was excited with a 445 nm diode laser (CUBE 445–40C, Coherent, Santa Clara, CA). Images were acquired at 128 × 128 pixels with 1 × 1 binning, in a 14-bit dynamic range, and with 10 ms exposure time. The fluorescence signal was captured by the imagers with 100 Hz through 578/105 bandpass filters (Semrock, Lake Forest, IL). The fluorescence change was defined as:

$$\Delta F/F_0 = (F_n - F_0)/F_0$$

where $F_n$ is the fluorescence at time point n, and $F_0$ is the average fluorescence before starting IR-laser irradiation (time window 100 ms). $F_{peak}$ is defined as the maximum amplitude of $\Delta F/F_0$. The fluorescence itself declined during IR-laser irradiation, so the decay was subtracted for quantification.

## Estimation of the onset timing of Ca²⁺ transients by curve fitting

Estimation of onset timings of Ca$^{2+}$ transients was performed as previously described (*Grewe et al., 2010*; *Lütcke et al., 2013*) with slight modifications:

$$f_{Ca}(t) = A\left(1 - e^{-(t-t_0)/\tau_{on}}\right) \cdot e^{-(t-t_0)/\tau_{off}}, \ \text{ for } t > t_0$$

$$f_{Ca}(t) = 0, \qquad\qquad \text{for } t \le t_0$$

Here, $t_0$ denotes the onset of $Ca^{2+}$ transients, $\tau_{on}$ the onset rise time constant, $\tau_{off}$ the decay time constant, and A an amplitude scale parameter. $\tau_{on}$ and $\tau_{off}$ were manually adjusted for precise curve fitting in each trial ($\tau_{on}$=50–500 ms, $\tau_{off}$ = 1.5–10 s). The value of A is dependent on the maximum of each $Ca^{2+}$ transient. Before fitting, a baseline offset was subtracted from the trace segment. We used MATLAB scripts provided in *Lütcke et al. (2013)*.

## $Cl^-$ imaging

$Cl^-$ imaging of SuperClomeleon-expressing Class IV neurons was performed on whole-mount preparations. The data acquisition system was the same as for $Ca^{2+}$ imaging of TN-XXL (*Terada et al., 2016*). The ratio of SuperClomeleon was defined as:

$$Ratio_{\text{SuperClomeleon}} = \left(YFP_{\text{unmasked}} - YFP_{\text{masked}}\right)/\left(CFP_{\text{unmasked}} - CFP_{\text{masked}}\right)$$

where $YFP_{\text{unmasked}}$ and $CFP_{\text{unmasked}}$ are signals of outlined cellular regions, and $YFP_{\text{masked}}$ and $CFP_{\text{masked}}$ are those of background.

## Thermal behavioral assay

Animals were raised at 25°C in an incubator with 12 hr light/dark cycles, and humidity was manually controlled (75–80%). Wandering third-instar larvae were gently picked up from the vial, washed three times with deionized water, and transferred to a 140 × 100 mm petri dish with fresh 2% agarose. Excessive water was removed from the animals. For acclimation, animals were allowed to rest on the plate for at least 5 min before testing. The response latency was measured as the time interval from the point at which the larva was first contacted by the probe until it initiated the first 360° rotation. The time window was 5 s when we could maintain the contact more precisely than the general time window (10 s). About 20 larvae in the control and experimental groups were tested on the same day, and the assays were repeated for several days.

## Image acquisition and quantification of dendritic morphology

Imaging ddaC neurons in whole-mount larvae was done as previously described (*Shimono et al., 2014*), with slight modifications. Wandering third-instar larvae were gently picked up from the vial, and washed once with 0.7% NaCl and 0.3% Triton X-100, and three times with deionized water. They were mounted in 50% glycerol on slides, between spacers made of vinyl tape. Images of YFP fluorescence in TN-XXL were acquired using a Nikon C1 laser-scanning confocal microscope. Original images of each neuron were composed of maximum intensity projections of confocal micrographs.

Dendritic coverage was quantified as previously described (*Honjo et al., 2016*) with appropriate modifications (*Figure 3—figure supplement 3A*). Original images were inverted with black and white, and the images were converted through a Laplacian filter to enhance the edge contrast and the VanderBrug operator (*Vanderbrug, 1976*; *VanderBrug, 1977*) to enhance the line contrast. The images were binarized to detect the enhanced parts. The images were converted through a maximum filter to interpolate between separated dendrites. Spotted noise was removed by labeling. The images were overlaid with a grid of 34 × 34 pixel squares (14 × 14 µm), and squares containing signals were counted to calculate the dendritic coverage score. The preceding quantification steps were automatically processed with a MATLAB script. After that, false-positives and false-negatives were manually corrected on a MATLAB application using a graphical user interface (GUI).

## Optogenetic neural activation

Optogenetic activation of Class IV neurons was performed as previously described (*Terada et al., 2016*). Larvae expressing the ChR2 variant in Class IV neurons driven by *TrpA1-QF (attP40)* (*Petersen and Stowers, 2011*; *Yoshino et al., 2017*) were grown on fly food containing all trans-retinal (R2500; Sigma-Aldrich) at a final concentration of 0.5 mM. For optogenetic activation, a single long pulse (continuous illumination) or multiple cycles of 100 ms pulses followed by 100 ms pause intervals (intermittent illumination) were applied by using a collimated LED light lamp (M470L3-C1; ThorLabs, Newton, NJ) with an emission peak at around 470 nm (0.37 mW/mm²). Each cell was stimulated by two illumination patterns temporally separated by a pause interval of at least 1.5 min. The first stimulus was a continuous pattern, and the second was an intermittent one in half of the trials;

and the sequence of stimulus patterns was reversed in the other half. We did not find any differences of neuronal activities dependent on the order of the two types of illuminations.

$Ca^{2+}$ imaging of jRCaMP-expressing Goro neurons was performed on optimized filet preparations as follows: (i) After basic preparations on a glass slide, anterior thoracic epidermis was cut off. (ii) Epidermis under the larval brain was slit vertically from the anterior side, and the brain was gently pinned on the slide. (iii) The samples were incubated with 7 mM monosodium glutamate (Nacalai, Kyoto, Japan) for 8 min to prevent muscle contractions and eliminate motor feedback to the sensory circuits by saturating glutamate receptors at the neuromuscular junction (*Kaneko et al., 2017*).

Imaging was done as previously described (*Arata et al., 2017*), with slight modifications. Data were collected on an IX71 microscope (Olympus) equipped with an objective (UPLSAPO60XS NA 1.3, Olympus), Nipkow disk confocal system (CSU10, Yokogawa Electric, Tokyo, Japan), and an EM-CCD camera (iXon$^{EM}$+ DU-888, Andor Technology, Belfast, UK). jRCaMP was excited with a 561 nm diode laser (Sapphire, Coherent, Santa Clara, CA), and was captured by the imagers at 1.5 s intervals through 610/60 bandpass filters (Chroma Technology, Bellows Falls, VT). The above imaging system was controlled by MetaMorph software (Molecular Devices, Sunnyvale, CA). Each sample was stimulated once either by continuous illumination or by intermittent illumination. Quantification of the fluorescence change was the same as for $Ca^{2+}$ imaging of GCaMP5G.

## Statistics

Data were analyzed and plotted using ImageJ (National Institutes of Health, Bethesda, MD), MATLAB (The MathWorks, Natick, MA), and Microsoft Excel (Microsoft Corporation, Redmond, WA). Details for each figure are shown in source data. To prevent misinterpretation as outliers in some figures (*Figure 2E and F*; *Figure 2—figure supplement 2*), $p < 0.05$ and $p < 0.01$ are indicated by double and triple asterisks, respectively.

## Abbreviations

$[Ca^{2+}]_i$, intracellular $Ca^{2+}$ concentration; $[Cl^-]_i$, intracellular $Cl^-$ concentration; GUI, graphical user interface; IR, infrared; ISI, interspike interval; $K_{2P}$, Two-pore domain $K^+$ channel; $K_{Ca}$, $Ca^{2+}$-activated $K^+$ channel; $K_{ir}$, Inward rectifier $K^+$ channel; $K_v$, voltage-gated $K^+$ channel; NR, no response group; ns, not significant; *ppk*, *pickpocket*; SK channel, small conductance $Ca^{2+}$-activated $K^+$ channel; US, unconventional spike; VGCC, voltage-gated $Ca^{2+}$ channel.

## Acknowledgements

Reagents, genomic datasets, and/or facilities were provided by the Department of *Drosophila* Genomics and Genetic Resources (DGGR) at Kyoto Institute of Technology, the Bloomington Stock Center, Vienna *Drosophila* Resource Center, the TRiP at Harvard Medical School (NIH/NIGMS R01-GM084947), the NIG stock center, the Developmental Studies Hybridoma Bank at the University of Iowa, the *Drosophila* Genomics Resource Center (DGRC), FlyBase, C Kim, and T Ohyama. An original MATLAB script for quantification of dendritic coverage was kindly provided by K Honjo and WD Tracey. We also thank J Hejna for polishing the manuscript; and K Oki, R Moriguchi, and M Futamata for their technical assistance. This work was supported by a grant from the programs Grants-in-Aid for Scientific Research on Innovative Areas' Mesoscopic Neurocircuitry' (22115006 to TUe), a grant from Takeda Science Foundation, the Platform Project for Supporting in Drug Discovery and Life Science Research (Platform for Dynamic Approaches to Living System) from Japan Agency for Medical Research and Development (AMED) to TUe, a Grant-in-Aid for Scientific Research (C) to TUs (24500410), a grant from the programs Grants-in-Aid for Scientific Research on Innovative Areas' Brain Environment' to TUs (24111525), and Toray Science and Technology Grant from Toray Science Foundation to TUs. KO is a recipient of a JSPS Research Fellowship for Young Scientists.

## Additional information

### Funding

| Funder | Grant reference number | Author |
|---|---|---|
| Japan Society for the Promotion of Science | | Koun Onodera |
| Ministry of Education, Culture, Sports, Science, and Technology | Grants-in-Aid for Scientific Research on Innovative Areas 'Mesoscopic Neurocircuitry' 22115006 | Tadashi Uemura |
| Takeda Science Foundation | | Tadashi Uemura |
| Ministry of Education, Culture, Sports, Science, and Technology | Grant-in-Aid for Scientific Research (C) 24500410 | Tadao Usui |
| Ministry of Education, Culture, Sports, Science, and Technology | Grants-in-Aid for Scientific Research on Innovative Areas 'Brain Environment' 24111525 | Tadao Usui |
| Toray Industries | Toray Science and Technology Grant from Toray Science Foundation | Tadao Usui |
| Ministry of Education, Culture, Sports, Science, and Technology | Platform Project for Supporting in Drug Discovery and Life Science Research from AMED | Tadashi Uemura |

The funders had no role in study design, data collection and interpretation, or the decision to submit the work for publication.

### Author contributions

Koun Onodera, Conceptualization, Resources, Data curation, Software, Formal analysis, Funding acquisition, Investigation, Visualization, Methodology, Writing—original draft, Writing—review and editing; Shumpei Baba, Conceptualization, Resources, Data curation, Formal analysis, Investigation, Methodology; Akira Murakami, Resources, Software, Writing—original draft; Tadashi Uemura, Conceptualization, Supervision, Funding acquisition, Methodology, Writing—original draft, Project administration, Writing—review and editing; Tadao Usui, Conceptualization, Resources, Supervision, Funding acquisition, Methodology, Writing—original draft, Project administration, Writing—review and editing

### Author ORCIDs

Koun Onodera (iD) http://orcid.org/0000-0002-4203-9865
Tadashi Uemura (iD) http://orcid.org/0000-0001-7204-3606
Tadao Usui (iD) http://orcid.org/0000-0002-0507-1495

### Decision letter and Author response

Decision letter https://doi.org/10.7554/eLife.29754.029
Author response https://doi.org/10.7554/eLife.29754.030

## Additional files

### Supplementary files

• Transparent reporting form
DOI: https://doi.org/10.7554/eLife.29754.027

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
