## [Decision Letter]

Thank you for submitting your article "Small conductance Ca^2+^-activated K^+^ channels induce the firing pause periods during the activation of *Drosophila* nociceptive neurons" for consideration by *eLife*. Your article has been reviewed by two peer reviewers, and the evaluation has been overseen by K VijayRaghavan as the Senior Editor and Reviewing Editor.

The reviewers have discussed the reviews with one another and the Reviewing Editor has drafted this decision to help you prepare a revised submission.

Summary:

In a previous study published in *eLife*, the authors' group reported that "burst and pause" activity pattern of Class IV multi-dendritic neurons in *Drosophila* larvae, accompanying a calcium transient, is induced by heat but not light. The authors also showed that a L-type voltage-gated calcium channel is required for the formation of calcium transients and proper behavioral response to noxious heat (rolling). However, molecular mechanisms that regulate the temporal pattern of the activity were unknown. In this manuscript, Terada et al. substantially extended this previous study by showing that the "burst and pause" activity is regulated in part by SK channels. SK channels are known to regulate a complex activity pattern in mammalian Purkinje cells, raising an interesting possibility that similar molecular strategies are utilized to generate fluctuated neural activity across species. I think that the results described in this manuscript represent substantial developments that merit publication as a Research Advance.

Essential revisions:

There are a few concerns that the authors should address before publication of the article.

1) The authors propose that calcium influx via L-type VGCCs activates the SK channels, which in turn generate the pause. However, the number of pauses is also increased when other potassium channels that are not dependent on calcium, such as Shaker, are knocked-down. How does this observation fit with the authors' model?

2) Based on the observation that the behavioral response to heat stimulation is enhanced in the SK knockdown, the authors propose that the burst-and-pause firing pattern provokes the behavioral response. However, the enhanced behavioral responses could be due to changes in other electrophysiological properties of the cells, unrelated to the burst-and-pause firing pattern. In this regard, the authors may study if the behavioral responses are also enhanced in other gene knockdowns (*Sh, Shal, Irk2*, and *Task7*) that show increase in the burst-and-pause firing patterns. If this were the case, it will provide strong evidence that the burst-and-pause firing pattern is responsible for the induction of the rolling behavior.

3) It would strengthen the study if the authors could show that some of the downstream neurons implicated in rolling behavior (e.g. Basin-4) responded differently to noxious heat stimuli when the fluctuation code is altered in SK and VGCC mutants.

---

## [Author Response]

Essential revisions:There are a few concerns that the authors should address before publication of the article.1) The authors propose that calcium influx via L-type VGCCs activates the SK channels, which in turn generate the pause. However, the number of pauses is also increased when other potassium channels that are not dependent on calcium, such as Shaker, are knocked-down. How does this observation fit with the authors' model?

We suggest that the other K^+^ channels, such as Shaker, also hyperpolarize the membrane potential to some degree after depolarization, regardless of dendritic Ca^2+^ influx, because most of them are voltage-dependent. Because the hyperpolarization suppresses the probability of firing including unconventional spikes (USs), the knockdown of those other channels should lead to the increment of the number of burst-and-pause patterns. To elaborate on this point, we corrected the description in the Discussion (second paragraph).

2) Based on the observation that the behavioral response to heat stimulation is enhanced in the SK knockdown, the authors propose that the burst-and-pause firing pattern provokes the behavioral response. However, the enhanced behavioral responses could be due to changes in other electrophysiological properties of the cells, unrelated to the burst-and-pause firing pattern. In this regard, the authors may study if the behavioral responses are also enhanced in other gene knockdowns (Sh, Shal, Irk2, and Task7) that show increase in the burst-and-pause firing patterns. If this were the case, it will provide strong evidence that the burst-and-pause firing pattern is responsible for the induction of the rolling behavior.

We examined heat avoidance behavior of larvae in which one of the other candidate genes was knocked down. Among the four suggested genes, we could not investigate the behavior of *Sh* knockdown larvae because they hardly developed into mature larvae. The knockdown larvae of the remaining three channel genes (*Shal, Irk2*, and *Task7*) did not show any difference in the response rate of avoidance (Figure 3—figure supplement 4). Although this result was at variance with the reviewers’ expectation, we did not think that it was inconsistent with our hypothesis. Interestingly, the frequencies of spontaneous spikes were significantly increased in these knockdown Class IV neurons, but not in the *SK* knockdown ones (Figure 3—figure supplement 4). Notably, a recent study revealed that the activation of Class IV neurons during larval development inhibited the synaptic transmission to second-order neurons via serotonergic feedback signaling and suppressed the avoidance behavior (Kaneko et al., 2017). In addition, the topographic projections of Class IV neurons are partially dependent on the levels of neuronal activity, including spontaneous spikes (Kaneko and Ye, 2015; Yang et al., 2014). Thus, the elevated basal neuronal activity during development might decrease the efficacy of synaptic transmission and/or remodel synaptic connections of the neurons, which counteract the effect of the increment of firing rate fluctuations on the avoidance behavior of these knockdown animals. So, we added the description in Results (subsection “SK channels generate pause periods”, third paragraph).

3) It would strengthen the study if the authors could show that some of the downstream neurons implicated in rolling behavior (e.g. Basin-4) responded differently to noxious heat stimuli when the fluctuation code is altered in SK and VGCC mutants.

Previously, we showed that distinct numbers of burst-and-pause firing patterns in Class IV neurons are induced by optogenetic activation (Terada et al., 2016). We considered that the firings with no burst-and-pause patterns in L-type VGCC mutants, which the reviewers had recommended, could be recapitulated by optogenetic activation by using continuous illumination; similarly, the firings with a large number of burst-and-pause patterns in *SK* mutants could be produced by using intermittent illumination. In designing these additional experiments, we estimated that it would take more time to generate fly strains required for the experiment with L-type VGCC or *SK* mutant backgrounds. Thus, we employed the optogenetic activation as an alternative method of thermal stimulation in the L-type VGCC and *SK* mutants (Figure 3—figure supplement 5).

We first investigated firing patterns induced by optogenetic activations in Class IV neurons, and we found that the numbers of burst-and-pause patterns were significantly different between continuous and intermittent illuminations (Figure 3—figure supplement 5; *P* < 0.001, Wilcoxon signed-rank sum test). Importantly, the total spike numbers and maximum firing rates were comparable between the two conditions (Figure 3—figure supplement 5). We then examined whether the activity of the downstream neurons was differentially induced with the type of optogenetic manipulation. We found that the maximum amplitude of Ca^2+^ rises in Goro neurons was larger with activation associated with more burst-and-pause firing patterns in Class IV neurons (Figure 3—figure supplement 5; *P* < 0.01, Welch’s *t*-test). The results suggested that firing rate fluctuations were decoded in downstream circuits separately from the firing rate itself. These findings strengthen our hypothesis that fluctuation coding occurs in the sensory mechanism of *Drosophila* larvae. Therefore, we added the description in the Introduction (last paragraph) and Results (last paragraph).

Shumpei Baba has been added as an author at the revised stage because he joined the experimental design for the revision and conducted the Ca^2+^ imaging experiment of Goro neurons (Figure 3—figure supplement 5). We have already confirmed that all authors agreed with their inclusion and place in the author list.